



# Machine learning models to replicate large-eddy simulations of air pollutant concentrations along boulevard-type streets

Moritz Lange[1], Henri Suominen[1], Mona Kurppa[2], Leena Järvi[2,3], Emilia Oikarinen[1], Rafael Savvides[1], and Kai Puolamäki[1,2]

[1]Department of Computer Science, University of Helsinki, Finland
[2]Institute of Atmospheric and Earth System Research (INAR) / Physics, Faculty of Science, University of Helsinki, Finland
[3]Helsinki Institute of Sustainability Science, Faculty of Science, University of Helsinki, Finland

**Correspondence:** Kai Puolamäki (kai.puolamaki@helsinki.fi)

**Abstract.** Running large-eddy simulations (LES) can be burdensome and computationally too expensive from the application point-of-view for example to support urban planning. In this study, regression models are used to replicate modelled air pollutant concentrations from LES in urban boulevards. We study the performance of regression models and discuss how to detect situations where the models are applied outside their training domain and their outputs cannot be trusted. Regression models

from 10 different model families are trained and a cross-validation methodology is used to evaluate their performance and to find the best set of features needed to reproduce the LES outputs. We also test the regression models on an independent testing dataset. Our results suggest that in general, log-linear regression gives the best and most robust performance on new independent data. It clearly outperforms the dummy model which would predict constant concentrations for all locations (mRMSE of 0.76 vs 1.78 of the dummy model). Furthermore, we demonstrate that it is possible to detect concept drift, i.e., situations where

the model is applied outside its training domain and a new LES run may be necessary to obtain reliable results. Regression models can be used to replace LES simulations in estimating air pollutant concentrations, unless higher accuracy is needed. In order to have reliable results it is however important to do the model and feature selection carefully to avoid over-fitting and to use methods to detect the concept drift.

## 1 Introduction

Exposure to ambient air pollution leads to cardiovascular and pulmonary diseases, and is estimated to cause 3 million premature deaths worldwide every year (Lelieveld et al., 2015; WHO, 2016), of which 0.8 million occur in Europe (Lelieveld et al., 2019). Urban areas are generally characterized not only by high population densities but also higher air pollutant concentration levels compared to rural areas. The degraded air quality particularly in street canyons results from high local emissions, such as traffic

combustion near the ground, as well as limited dispersion of these traffic-related pollutants. Streets with traffic are generally flanked with buildings and/or vegetation that inhibit pollutant ventilation upwards from the pedestrian level. Namely, dispersion



is the main factor determining air quality (e.g., Kumar et al., 2011). Furthermore, these obstacles block, decelerate and modify air flow, leading to a highly turbulent wind field and pollutant dispersion patterns (Britter and Hanna, 2003). Consequently, certain urban planning solutions can be applied to enhance air pollutant dispersion and hence to improve local air quality to

some extent (see, e.g., Kurppa et al., 2018; Nosek et al., 2016; Yuan et al., 2014). This, on the other hand, evokes the need for high-resolution, building-resolving air pollution modelling.

Successful modelling of urban air pollutant dispersion necessitates taking into account the detailed properties of adjacent buildings and vegetation in the area of interest as well as in its surroundings. To date, high-resolution dispersion modelling has mainly been based on physical modelling techniques, of which computational fluid dynamics (CFD) models, notably

Reynolds-averaged Navier-Stokes equations (RANS) and large-eddy simulation (LES), are the most applicable tools for the purpose. CFD models solve the flow and dispersion around individual buildings, and with constantly increasing computational resources the modeling domains can currently be extended to cover entire neighbourhoods and even cities (Auvinen et al., 2020). However, conducting reliable and high quality CFD simulations requires expertise in model application. Moreover, for building-resolving simulations that apply a grid resolution of 2 m or finer, especially LES necessitates supercomputing

resources. At the same time, LES has been found more accurate than RANS in solving finer scale details (Salim et al., 2011; Tominaga and Stathopoulos, 2011) and therefore particularly suitable in modelling flow and pollutant concentrations within real urban environments. As a contrast to CFD, statistical models based on machine learning may offer a significantly less expensive alternative to predict urban air quality and pollutant dispersion. Consequently, the number of studies conducting machine-learning based air quality modelling has increased rapidly (Rybarczyk and Zalakeviciute, 2018).

Machine learning allows finding a relationship between a target variable, e.g., the concentration of air pollutants in a certain location, and its predictors, which are often called features. These types of machine learning models are called regression models. The models are trained on a specified training data which by some rule, e.g., maximising the likelihood given that the relationship is linear with normally distributed noise, observe a relationship between the target variable and its features. To evaluate the trustworthiness of the results, often a part of the available data is not used to train the model but to evaluate it.

This 'unseen' evaluation data gives an estimate of the model performance in a realistic urban planning scenario. Perhaps the largest advantage of regression models compared to the CFD models is their speed. However, the increase in speed comes at a cost of accuracy. Another disadvantage is that accurate predictions require the predicted data to follow approximately the same distribution as the training data, reducing their use to only similar modelling set-ups as they have been trained in.

Most of the previous studies on developing a statistical air pollution model using machine learning have been based on

field measurements, and the spatio-temporal distribution of pollutants has been assessed by utilizing multiple stationary sites in model training (e.g., Araki et al., 2018; Yang et al., 2018). To further improve the spatial resolution of modelling in urban areas, also mobile air quality measurements have been utilised (e.g., Adams and Kanaroglou, 2016; Hu et al., 2017; Krecl et al., 2019; Van den Bossche et al., 2018). However, due to spatial accuracy constraints, the spatial resolution applied has been limited to the order of 10–50 m, which does not allow investigating the impact of individual buildings on pollutant dispersion.

Output data from a numerical model have been used for training but only in the regional scale with a 5–15 km resolution (Feng



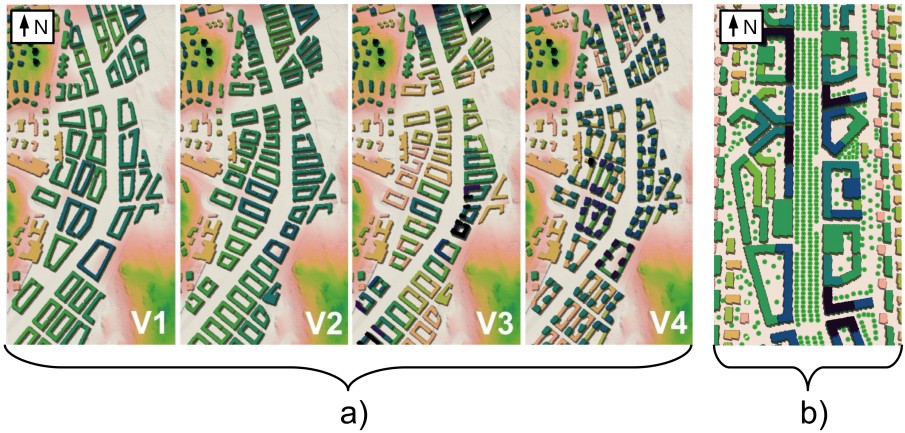

**Figure 1.** Simulation domains of the LES output data applied in the study: a) Four city-planning alternatives V1–4 investigated in KU18 (Kurppa et al., 2018), and b) City-boulevard scenario S1 and its surroundings studied in KA20 (Karttunen et al., 2020). Green dots illustrate trees. The city boulevard is 54 m and 58 m wide in KU18 and KA20, respectively.

et al., 2019; Peng et al., 2017). Machine learning studies applying LES data have so far been mainly restricted to turbulence closure modelling (e.g., King et al., 2018).

In this study, the application of machine learning for emulating LES outputs of local scale air pollutant dispersion in urban areas is investigated. We use LES outputs from two different studies conducted in different boulevard-type street canyons in Helsinki. Specifically, we create appropriate features for the neighbourhoods around the street canyons from the inputs of the LES that are used to train the machine learning models, and then evaluate the performance and reliability of different algorithms. The motivation is to approximate the computationally expensive simulations with machine learning models that are faster to evaluate. The ultimate goal is to develop a model that can easily be applied to support urban planning.

This study is structured as follows: First, the LES datasets and feature construction are described. Then brief descriptions of the used machine learning models and the training and evaluation process are provided. Finally, the applications, limitations, and future work are discussed.

## 2   Methods and material

In this chapter, we introduce the LES datasets used in this study and explain our pre-processing steps. Then, we present the features available to the regression models. Finally, we describe how the optimal set of features is chosen by forward feature selection and describe the used performance measures. All analyses were carried out in R version 3.6.2 (R Core Team, 2020).





## 2.1 Large-eddy simulation datasets

LES models resolve the three-dimensional prognostic equations for momentum and scalar variables. In LES, all turbulence scales larger than a chosen filter width are resolved directly. The smaller scales, which should represent less than 10% of the turbulence energy (Heus et al., 2010), are parametrised using a sub-grid scale model. This study uses output data from two
studies to train and evaluate the regression methods: Kurppa et al. (2018) and Karttunen et al. (2020). Both studies apply the LES model PALM (Maronga et al., 2015, 2019) to assess the impact of city planning on the pedestrian-level air quality.

The first study by Kurppa et al. (2018, model revision 1904, hereafter KU18) investigates the impact of city-block orientation and variation in the building height and shape on the dispersion of traffic-related air pollutants. Specifically, LES is run over a 54-m-wide city boulevard applying four alternative city-planning solutions (V1-V4 in Fig. 1a). Here, simulations with a
neutral atmospheric stratification and wind from east (90°) and south-west (225°) are considered. Air pollutant dispersion is studied by a Lagrangian particle model embedded in PALM. Air pollutants are represented as inert particles that are released above streets with traffic and follow the air flow without interacting with any surface. In the present study, 40-minute averaged concentration fields from KU18 are employed.

The second study by Karttunen et al. (2020, model revision 3698, hereafter KA20) assesses the impact of street-tree layout
on the concentrations of traffic-related aerosol particles. The study is conducted over a 50 to 58-m-wide city boulevard (Fig. 1b) with neutral atmospheric stratification under the two wind directions: parallel and perpendicular to the boulevard. Contrary to KU18, aerosol particle concentrations and size distributions as well as aerosol dry deposition on surfaces and vegetation are explicitly modelled by applying an aerosol module embedded in PALM (Kurppa et al., 2019). The aerosol module applies the Eulerian approach. Here, one-hour averaged concentration fields of $PM_{2.5}$ (particulate matter with aerodynamic diameter $<$
2.5 μm) for one modelling scenario (S1) are applied.

Both studies apply a grid spacing of 1.0 m in horizontal and 0.75–1.0 m in vertical to directly resolve the most relevant turbulent structures related to buildings and vegetation. The surface description is given to PALM by maps: namely those of topography elevation, tree height and emission strength per area. The simulations were conducted in a supercomputing environment and each simulation took approximately $10^3$ days of CPU time.

The machine learning models in this study are trained with KU18 and evaluated with KA20. Using KA20 for evaluation mimics a realistic urban planning scenario, where KA20 would correspond to a new city plan considered by the urban planner. KU18 is selected as the training dataset due to greater variety in building layouts compared to KA20, in which the variation is mainly limited to different street-tree scenarios. Clear differences in the building layouts lead to deviant pollutant dispersion and concentration distributions, which improves the generalisation performance. An alternative approach for training would
entail creating a training and evaluation data using random samples from both KU18 and KA20. However, random sampling is impractical in this case due to significant differences between KU18 and KA20. KU18 simulates dispersion qualitatively and assumes weightless and inert particles that imitate air pollutants in general. In contrast, KA20 models realistic aerosol particle concentration values and includes realistic model physics for the aerosol dry deposition. These differences affect the scaling and distributions of the simulated particle concentrations (Figure 2), which is further discussed in Sections 3.2 and 3.3.



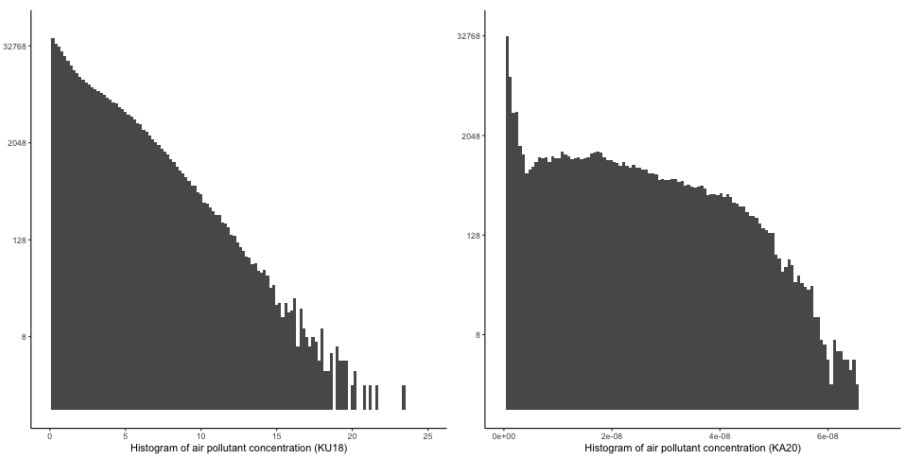

**Figure 2.** Histograms of air pollutant concentrations in KU18 and KA20. Note the log-scale of the counts on the $y$-axis, and the differing scales on the $x$-axis. KA20 has units $\mathrm{kg\,m^{-3}}$.

## 2.2 Data pre-processing

The LES outputs KU18 and KA20 are pre-processed into a suitable format for training the regression models. The pre-processing is divided into two parts: aggregating the target variable over time and height, and constructing expressive features from the LES inputs.

### 2.2.1 Target variable

In regression, the target variable is predicted using predictor variables (features). Here, the target variable is the (air) pollutant concentration $pc$. The raw LES outputs contain $pc$ on a spatio-temporal grid $(x, y, z, t)$. We pre-process the target variable by averaging $pc$ over time and considering a fixed height (at which pollutants are emitted). Then, $pc$ are predicted on the $(x, y)$ grid instead of the $(x, y, z, t)$ grid. This simplifies the modelling task significantly while still retaining the relevant information on $pc$ at the pedestrian level.

In addition to time-averaging, we also restrict the spatial extent in which the regression models are trained. The area of interest in this study is the city boulevard in the middle of the maps, which is the same as in the original studies that generated KU18 and KA20. In KA20, the boulevard is surrounded by artificial buildings to imitate the aerodynamic roughness of an suburban environment, which however are of no interest and are therefore omitted in the model development. Figure 3 presents the exact areas used for modeling. The spatio-temporal aggregate is calculated by temporally averaging the LES data for the first 100 seconds displaying stable behaviour. Furthermore, only one vertical level is considered: 4 m and 0.88 m above ground level for KU18 and KA20, respectively. This exact vertical level is chosen since it is the one at which the studied air pollutants are released in their respective simulations. In order to make the datasets more comparable, the background concentrations in KA20 are removed using the transformation $pc_{\mathrm{new}} = pc_{\mathrm{old}} - \min(pc_{\mathrm{old}})$.





**Table 1.** List of features used in the regression models.

| Feature name | Description |
| --- | --- |
| Building height | Height of the closest building |
| Canopy height | Height of the vegetation canopy |
| Courtyard | Binary variable indicating presence of a courtyard |
| Direction of closest building | Direction to the closest building relative to the wind direction |
| Distance to building downwind | Distance to closest building in the same direction as the wind |
| Distance to building upwind | Distance to closest building in the direction against the wind |
| Height to width ratio | Height of the closest building relative to the width of the street |
| Pollutant emissions | Emission level |
| Pollutant emissions convolution, $\sigma = \{1, 2, 4, 8, 16\}$ | Gaussian convolution of the emissions with standard deviation $\sigma$ |
| Pollutant emissions convolution upwind, $\sigma = \{8, 16, 32\}$ | Convolution of the emissions upwind with standard deviation $\sigma$ |
| Street | Binary variable indicating presence of a street |
| Street width | Width of the street |

### 2.2.2 Features

125 The target variable is predicted using features that are defined for each $1 \text{ m} \times 1 \text{ m}$ surface pixel in the modelling domain. The features include direct inputs to the LES and *surrogate features* which are constructed from these inputs. The direct LES inputs are 'Height of the topography', which includes solid obstacles such as buildings, 'Height of the canopy' and 'Amount of pollutant emissions'. The features used for training the regression models are described briefly in Table 1 and in more detail in Table A1.

130 The surrogate features provide information about spatial dependencies in the modelling domain, reducing the need for the regression models to explicitly model these dependencies. For instance, applying convolutions of various sizes on pollutant emissions creates surrogate features of average pollutant emission densities over a spatial neighbourhood, weighted by proximity to the point in question. Other features, such as the height-to-width ratio of a street canyon, are created based on domain knowledge. Incorporating domain knowledge is important, since well-crafted input features largely determine the quality of 135 modelling results. Furthermore, understandable features aid experts in interpreting the physical meaning of the results.

### 2.3 Forward feature selection

After calculating all potentially useful input features, a subset of features to be used with each model is selected using forward selection (Hastie et al., 2009). Forward selection is a feature selection algorithm in which a model is trained iteratively with progressively larger feature subsets. Initially, every feature is used as a single predictor to train the model, and the best performing 140 feature is selected. The model is then re-trained using this feature along with every other feature as a second predictor, and the best performing second feature is selected. This process is repeated until either all features are selected or no additional feature




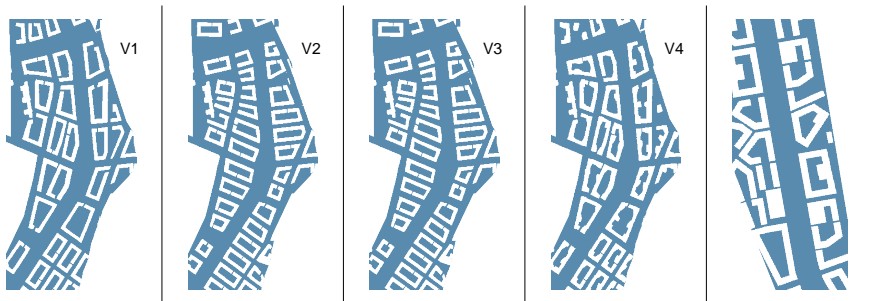

**Figure 3.** The map area cutout for which pollutant concentrations are modelled (in blue). For each of the city plans V1-4 from Kurppa et al. (2018) and for the map area from Karttunen et al. (2020) (rightmost picture).

improves the model. Forward selection limits the search space of all possible feature combinations considerably, and while it does not guarantee to find the globally best subset of features, it finds a local optimum. An advantage of forward selection that is relevant to this study is that it avoids selecting strongly correlated predictors, such as the different-sized convolutions of pollutant emissions (Table 1).

The best performing features are selected according to a selection criterion. Here the selection criterion is the cross-validation root mean squared error (RMSE). Cross-validation is a technique for evaluating generalisation performance of statistical models and for detecting over-fitting. In cross-validation, the data are split into $k$ random subsets, out of which $k-1$ are used to train the model, and one is used to validate the model. Due to the spatial auto-correlation in the LES data, a random split by sampling data points from all maps does not lead to statistically independent subsets. In order to ensure maximal independence between the training and validation data, the random split is performed at the level of city blocks. Each cross-validation split is trained with three different city plans under a given wind direction and validated with the fourth city plan under the other wind direction. This is repeated for all four city plans using both wind directions. As an example, one split uses city plans V1, V2 and V3 with the wind direction $90°$ as training data, and V4 with the wind direction $225°$ as validation data to validate predictions. Using each city plan for a given wind direction as unique validation data results in eight splits. The aggregated cross-validation error is then calculated as the error of all combined predictions of all splits.

## 2.4 Model descriptions

The applicability of ten common regression models trained on KU18 is examined, from the simplest linear model to the powerful support vector regression (SVR) model (Table 2).





**Table 2.** Models and their implementation in R (version 3.6.2.). All but two model pollutant concentrations ($pc$). For log-linear regression and logarithmic support vector regression, the model internally estimates $\log(pc+1)$, where $+1$ is added because of the zero values contained within $pc$. The predictions are transformed back to the original scale after being computed.

| Model name | Description | Implementation |
|---|---|---|
| Decision tree | Hierarchical model separating data based on rules | rpart |
| Gaussian process | A Bayesian kernel-based method for regression | kernlab |
| Gradient boosting | Ensemble method of decision trees | xgboost |
| Linear regression | Ordinary least squares linear regression | lm |
| Log-linear regression | Linear regression modelling $\log(pc+1)$ | lm |
| Poisson regression | Linear regression assuming Poisson distributed data | glm |
| Random forest | Ensemble method of decision trees | randomForest |
| Support vector regression | Non-linear kernel-based regression method | e1071 |
| Logarithmic support vector regression | Support vector regression modelling $\log(pc+1)$ | e1071 |
| Zero-inflated Poisson regression | Combination of logistic regression and Poisson regression | pscl |

**Linear models** Four generalised linear models are considered: linear regression, log-linear regression (Benoit, 2011), Poisson regression, and zero-inflated Poisson regression (Lambert, 1992). Generalised linear models model the target variable as a function of linear combinations of features. Linear models are relatively simple, which limits their flexibility, but makes them more interpretable. For example, if the features are normalised, then the regression coefficient of a feature communicates how much a change of one unit affects the mean of the target variable, given that all other features are

constant.

  **Tree-based models** Three tree-based models are considered: decision trees, random forest, gradient boosting with decision trees (Hastie et al., 2009). Decision trees model the target variable using simple if-else rules, which makes them interpretable. Random forest is an ensemble method that aggregates the predictions of multiple decision trees trained on random subsets of data and features. Gradient boosting is also an ensemble method, in which multiple decision trees

are trained sequentially on the results of previous trees, correcting their weaknesses. Ensemble methods achieve high prediction accuracy by pooling the predictions of multiple models. In particular, the random forest is among the best performing regression models (Fernández-Delgado et al., 2014). However, the complexity of ensemble models means that they are not interpretable.

  **Support vector regression** (SVR, Cristianini and Shawe-Taylor, 2000) is a powerful regression method that implicitly trans-

forms the data into a higher dimensional feature space. This enables SVR to model interactions and conditional dependencies between features, and hence utilize more information compared to simpler models. However, as with the complex tree models, the complexity of SVR leads to difficulties in understanding how the relations in the data are utilized by the model to estimate $pc$.





In addition to standard SVR, a log-transformed SVR is also used to enforce positive predictions for $pc$.

**Gaussian process regression** (GPR, Murphy, 2012) is a non-parametric approach to regression. GPR is a Bayesian method that uses a Gaussian process with a known covariance as a prior to infer the posterior predictive distribution of the unobserved values. It can be considered as an Bayesian alternative for other kernel methods, such as the SVR (Murphy, 2012). We use a squared exponential kernel as the covariance matrix for the model. This results in points having similar predicted values if they are close to each other in the feature space. The generally good performance of the GPR is also

complemented by its built-in ability to account for uncertainty.

**Dummy model** A dummy model is used as a baseline model for reference. The dummy model simply predicts mean $pc$ of the training data, regardless of any features.

## 2.5 Performance measure

The root mean squared error (RMSE) is a standard performance measure for model evaluation (Rybarczyk and Zalakeviciute, 190 2018). Here, a modified version of RMSE is used, due to the different scales of KU18 and KA20. We define the *multiplicative minimum-RMSE* based on a linear transformation of the predictions as $\mathrm{mRMSE}(pc, \hat{pc}) = \min_{a \in \mathbb{R}} \mathrm{RMSE}(a \cdot pc, \hat{pc})$ where $pc$ is a vector of the observed pollutant concentrations, and $\hat{pc}$ is a vector with the corresponding predictions. Using mRMSE is equivalent to using RMSE after scaling pollutant concentrations with a multiplicative factor $a$ that minimises the RMSE.

In addition to mRMSE, two other performance measures are presented in the results tables. The *cross-validation RMSE* 195 (cv-RMSE, described in Section 2.3) is presented as a performance measure on the training data, when using the optimal set of features selected with forward selection. *Scaled bias* is defined as $\mathrm{mean}(a \cdot pc - \hat{pc})$, where $a$ is the same multiplicative factor as in mRMSE. Scaled bias shows whether models over- or underestimate the pollutant concentrations.

Note that model performance on spatial data cannot be meaningfully summarised into a single performance measure. The spatial distribution of prediction errors can be examined using residual plots (Figure 5).

## 3 Experiments

This chapter describes the training process of the models with features selected using forward selection. Based on the training results, the best performing regression models are selected and summarized. The models are subsequently evaluated on an independent dataset (KA20). Lastly, the reliability of model predictions is assessed for cases when LES results are not available with a concept drift detection algorithm.

### 3.1 Model training

For each model, forward feature selection on KU18 provides the optimal set of features (as described in Section 2.3). Figure 4 shows the results of feature selection. Features are iteratively added ($x$-axis) and cv-RMSE is computed ($y$-axis). For each model, the features are selected such that cv-RMSE is minimized. In order to limit the computation time of feature selection,



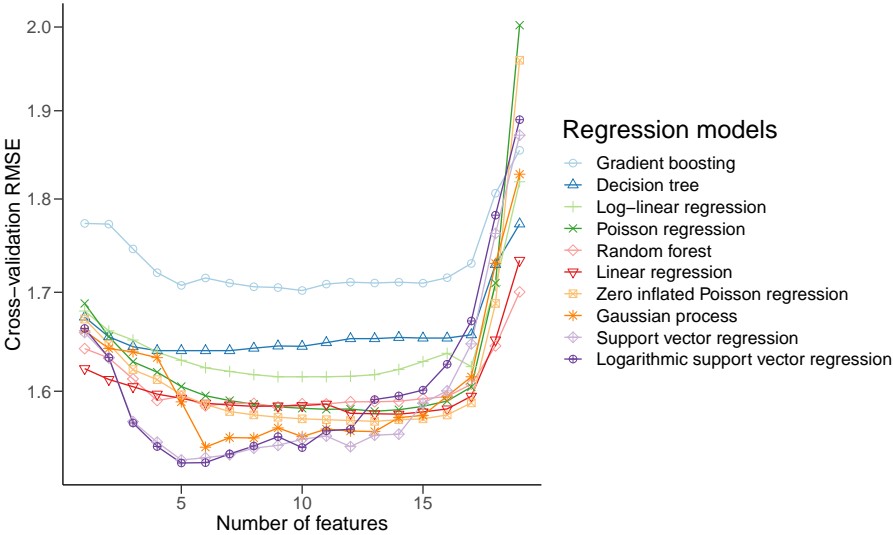

**Figure 4.** Feature selection results on KU18 for all regression models.

SVR and GPR were trained on a random subset of 2,500 data points (out of the total 472,991), rather than the whole cross-
validation split.

After feature selection, each model is trained on the whole training data (all KU18 city plans), using the optimal features
from feature selection. SVR and GPR are, as in feature selection, only trained on 2,500 randomly selected data points.

SVR and GPR have additional parameters, whose optimal values were computed using grid search. Grid search is a model
tuning technique in which the model is trained using all parameter values on a discrete grid of the parameter space. The
selection criterion for the optimal parameter values was cv-RMSE (as with feature selection).

Note that some models allow for negative predictions (negative pollutant concentrations), which is physically impossible.
One approach for avoiding negative predictions is to clip model outputs to a minimum of zero after prediction. For the purposes
of this study, however, negative predictions are retained, since their magnitude is relevant for error estimation.

## 3.2 Model selection

Out of the ten models described in Section 2.4, three are selected as the best performing for replicating the LES outputs:
logarithmic support vector regression, Gaussian process regression, and log-linear regression. Table 3 compares the three
selected models. Table 4 lists the performance of all ten models with respect to cross-validation RMSE, mRMSE, and scaled
bias (described in Section 2.5). Performance evaluation at this stage is based on the cross-validation errors computed during
forward feature selection on KU18. Final model evaluation on an independent dataset (KA20) is performed in Section 3.3.

The three best performing models are:





**Table 3.** Comparison of selected models based on KU18. Performance refers to the cross-validated RMSE, number of features refers to the optimal number selected during forward feature selection. Prior use means the model has been used for similar tasks in literature. A "+" means better than average, which is not necessarily more than average.

| Model | Performance | Number of features | Interpretable | Positive predictions | Prior use |
|---|---|---|---|---|---|
| Logarithmic SVR | + | + | × | ✓ | × |
| Gaussian process | + | + | × | × | ✓ |
| Log-linear regression | − | − | ✓ | ✓ | × |

**Logarithmic SVR** The logarithmic SVR has the smallest cross-validation RMSE, and as such, its predictions are the most accurate on the training data. Additionally, it only requires a small number of features to achieve this strong performance. Requiring low dimensional training data means that the user will need to provide fewer features with their data, minimising the expense of data preparation. A third advantage is that the log transformation ensures non-negative predictions for $pc$. Although the standard SVR does not ensure non-negative predictions, it requires the same number of features and offers almost the same RMSE.

**Gaussian process regression** The RMSE of the GPR is close to that of the log-SVR, making it one of the strongest models as well. In addition, it has previously been used to predict simulator outputs (Gómez-Dans et al., 2016).

**Log-linear regression** As a linear model, the log-linear regression is useful if model interpretability is required (for more details on its interpretability, see Section 2.4). What separates the log-linear regression from the other linear models is that it uses the fewest number of features, while also ensuring positive predictions through log-transformation, similar to the logarithmic SVR. It works as a simpler counterpart to the more powerful methods.

## 3.3 Model evaluation

As a final test, the models are evaluated on an independent dataset to obtain a more accurate estimate of their performance in a real-world urban planning situation. The models cannot be evaluated solely based on the training data, since model overfitting would not be detected, and the error estimation in real-world situations would be poor. For the evaluation, we use the models trained on both wind directions and all city plans of KU18 (as described in Section 3.1) and we evaluate them on both wind directions of KA20.

We use the mRMSE and the scaled bias defined in Section 2.5 for KA20, with the results for $PM_{2.5}$ concentrations listed in Table 4. Due to the scaling, the mRMSE does not allow direct comparisons between the errors on KU18 and KA20. On the evaluation data, we can however compare the models to the dummy model and see that, with the exception of the Poisson regressions, all models clearly outperform the dummy model, although the performance is more varied than with the training





**Table 4.** Cross validation and evaluation error for all models, obtained with their respective optimal set of features. Cross-validation RMSE is on KU18, while mRMSE and scaled bias are calculated on $PM_{2.5}$ in KA20. Highlighted are the three models that were ultimately selected for performance (based on cross-validation RMSE) or interpretability.

| Model | Number of features | Cross-validation RMSE | mRMSE ($PM_{2.5}$) | Scaled bias ($PM_{2.5}$) |
|---|---|---|---|---|
| Dummy model | 0 | 2.3 | 1.78 | -1.47 |
| Decision tree | 4 | 1.64 | 1.35 | -0.60 |
| **Gaussian process** | **6** | **1.55** | **0.91** | **-0.50** |
| Gradient boosting | 10 | 1.71 | 0.84 | -0.47 |
| Linear regression | 14 | 1.58 | 1.11 | -0.47 |
| **Log-linear regression** | **10** | **1.61** | **0.76** | **-0.47** |
| **Logarithmic support vector regression** | **5** | **1.53** | **0.87** | **-0.42** |
| Poisson regression | 13 | 1.58 | 2.12 | -0.70 |
| Random forest | 9 | 1.58 | 1.06 | -0.60 |
| Support vector regression | 5 | 1.53 | 0.83 | -0.40 |
| Zero-inflated Poisson regression | 13 | 1.57 | 1.80 | -0.65 |

data. The best performing model is the log-linear model (mRMSE = 0.76). The log-SVM and GPR also performed well (mRMSE = 0.87 and mRMSE = 0.91, respectively) when compared to the dummy model (mRMSE = 1.78).

Out of the selected models, the log-linear model is preferrable. Not only does it perform the best out of all the three models selected but it is also the simplest and it runs the fastest. Its simplicity is likely the reason for its good performance given the notable differences between the training and evaluation data. Although log-SVM and GPR both perform better than the average model, compared to the other models, their performance is worse in the evaluation data when comparing to the cross-validation procedure. This could be a sign of slight over-fitting to the training data or sensitivity of the different distribution of data on the

testing dataset.

The interpretation of the scaled bias is not straightforward due to the scaling, but it shows that all of the models overestimate $pc$ in the evaluation data with the SVR overestimating the least. The dummy model is, unsurprisingly, the most biased.

The scaled residuals can be seen in Figure 5 for the selected models. At a glance they may seem to be similar for different models but there are differences between the predictions displayed even by the different mRMSE. The differences are due to

the dissimilar ways of building a model and the features selected in Section 2.3. All three selected models acquire most of the error on the boulevard, while the outskirts are predicted more consistently. This is not a surprise since $pc$ is much lower outside of the boulevard. The figure also show that the log-linear model has more balanced residuals while the more complex SVR and GPR are able to achieve lower residuals on the outskirts but perform relatively poorly on the boulevard. The residuals also show that none of the models are fully capable of capturing details in the pollutant dispersion that arise due to fine-scale flow

patterns.

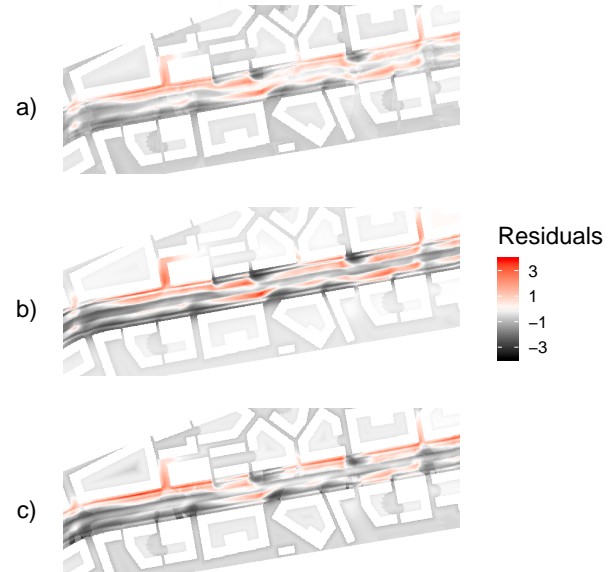

**Figure 5.** Model residuals of PM$_{2.5}$ predictions in KA20 for a) the log-linear regression, b) the logarithmic SVR and c) the Gaussian process. The wind is coming from the left. Residuals are calculated after scaling with the respective optimal $a$ for mRMSE calculation. Notice that although the residuals look similar, the predictions have notable differences. The scaled $pc$ ranges up to a maximum of 7.28. The residuals can also be contrasted to the mean squared error of the dummy model, which is 1.78.

## 3.4 Concept drift detection

It is important to know whether the results obtained generalize to different city plans. Because of the high computational cost of running LES, we often do not have access to the LES output values of a new plan, and hence we cannot assess the prediction error directly in such a case. We can, however, use the Drifter algorithm by Tiittanen et al. (2019) to estimate whether RMSE of a model prediction is high or not. The Drifter algorithm is designed for detecting *virtual concept drift* (Gama et al., 2014), i.e., detecting changes in the distribution of the features that affect the performance of the model. The idea behind Drifter is that we define a distance measure $d(x)$ that measures how far a covariate vector $x$ is from the data that has been used to train the model. Small values of $d(x)$, which is called the *concept drift indicator*, mean that we are close to the training data and the model should be reliable, while a large value of $d(x)$ means that we have moved away from the training data, after which the regression estimate may be inaccurate. The distance measure $d(x)$ is defined as follows. First, a family of so-called *segment models* $f_i$ is trained using only a part of the training data for each. Then, for each of the segment models $f_i$, we can compute an estimate of the generalization error using the terms $(f(x) - f_i(x))^2$ instead of the terms $(f(x) - pc)^2$, where $pc$ is the LES output value, when computing the RMSE. For a simple linear model, this kind of a measure is monotonically related to the expected quadratic error of the model (Tiittanen et al., 2019). Now, the ensemble of estimates for the segment models allows





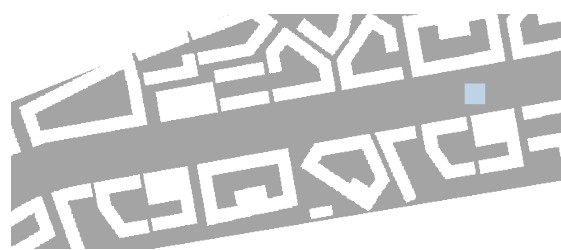

**Figure 6.** Example of a single evaluation segment (light blue square) applied in the concept Drift analysis for KA20 using the Drifter algorithm.

to compute a statistic for estimating RMSE. In Tiittanen et al. (2019) the statistic (i.e., the concept drift indicator $d(x)$) was chosen to be the second smallest error estimate.

Originally Drifter has been used for time series data. We adapt it for spatial data here by selecting the *segments of the training data* not to be temporally close-by data points, but *spatially close-by points*. We hence divide the map into $k \times k$ squares that are used as the segments. An example of such a segment can be seen in Fig. 6. Drifter is run for all models considered in Section 3.2 as the complete model. The original article suggests the usage of a simple linear model as the segment model, but here a ridge regression with logarithmic transformation and with a small regularization parameter of $\lambda = 0.01$ is chosen for practical reasons. This gives similar results to a log-linear model while also being well-defined on the segments where some features are constant.

In order to keep the results comparable with Section 3, we also scale the evaluation dataset (KA20) by multiplying it with the constant that minimizes RMSE. This, if the evaluation data is not segmented, is equivalent to using the same multiplicative minimum-RMSE error measure as in Section 3. This decision does not affect the concept drift indicator but it will make the simulation error of the training and evaluation more comparable. We train Drifter using all eight simulation set-ups of KU18 (i.e., four city plans and two wind directions) and evaluate it on both wind directions of KA20. We use $100\,\mathrm{m} \times 100\,\mathrm{m}$ squares as our segments in the training data with each square overlapping $25\%$ out of four other training segments, and $25\,\mathrm{m} \times 25\,\mathrm{m}$ squares in the evaluation data with no overlap. The concept drift indicator $d(x)$ is chosen to be the tenth smallest error estimate. The overlapping scheme used is the same as in the original article, but since the data is two dimensional, it is applied for both



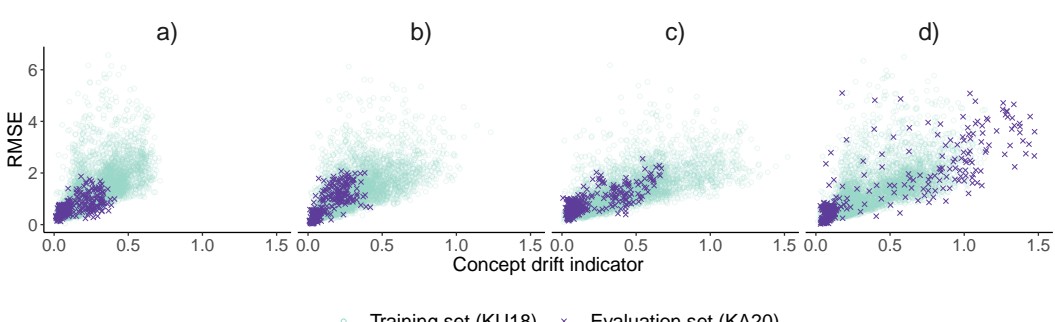

**Figure 7.** The relationship between the concept drift indicator and observed RMSE in a given segment in KU18 (circles) and KA20 (crosses) with a) log-linear regression , b) logarithmic support vector regression, c) Gaussian process, and d) log-linear regression with all features used as the complete model. The opacity of the training data has been reduced by $80\%$ to make the figure clearer.

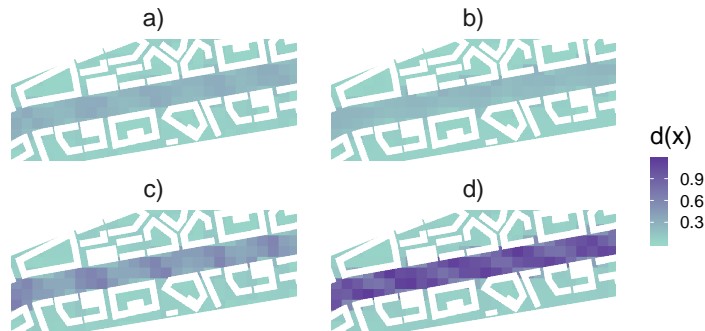

**Figure 8.** A map showing the concept drift indicator $d(x)$ in a given segment of KA20 for a) log-linear regression , b) logarithmic support vector regression, c) Gaussian process, and d) log-linear regression with all features used as the complete model. Notice how the last model has high concept drift indicator on the boulevard.

dimensions $(50\% \cdot 50\% = 25\%)$. These parameters are chosen with a grid search for the log-linear model with all features (a situation exhibiting concept drift).

In Figure 7a-c we show the concept drift indicator value and RMSE for each $25\,\mathrm{m} \times 25\,\mathrm{m}$ segment in the evaluation data
alongside the same sized segments in the training data for our final models in the same figure. With all three models considered, the evaluation data concept drift indicator is indeed correlated with its RMSE and thus can be used to estimate it. We also notice that for all three models the evaluation segments lie in the same area as the training segments indicating the lack of concept drift. In addition, we notice that the concept drift indicator values are larger on the boulevard which corresponds to the fact that the RMSE is indeed larger there. Therefore, the concept drift indicator is useful in detecting areas of large RMSE even when if
the ground truth (LES output) are not known.





Another example of the concept drift detection is given by testing Drifter with a sub-optimally performing model. If the same procedure is done for the log-linear model with all of the surrogate features and not with the set selected by forward selection, the model overfits. Unlike in the models above, we see from Figure 7d that multiple evaluation segments lie on the right of the cluster of the training segments indicating concept drift. These points also have a higher RMSE which shows that Drifter is
working as intended. From Figure 8d we see that concept drift is detected only on the boulevard which is as expected, because also the RMSE are higher there.

## 4   Discussion and conclusions

This study demonstrates that machine learning methods trained with LES data can be used to model street-level pollutant concentrations in a city-boulevard-type urban neighbourhood. The accuracy of the models is explored with an independent
evaluation dataset to ensure their applicability in urban planning for new, similar types of neighbourhoods.

The log-linear regression has the greatest potential for replicating LES results even with a relatively small amount of data. It also has much potential in helping to understand which urban features govern local pollutant concentrations. The kernelized methods, SVR and GPR, show moderate performance and perform generally well in capturing the local mean concentration. However, all three have trouble in representing smaller scale details in the concentration fields linked to turbulence. Further-
more, all models perform worse on the boulevard than in its surroundings. Still, these models beat the dummy model by a notable margin (e.g., RMSE $\leq 0.91$ for the selected models compared to RMSE $= 1.78$ for the dummy model). In general, bias in the models is slightly negative.

No previous studies on applying LES air pollution data to train a machine learning model exist and therefore direct comparison is unfeasible. Also comparing to studies applying spatial air quality measurements (Adams and Kanaroglou, 2016; Hu
et al., 2017; Krecl et al., 2019; Van den Bossche et al., 2018) is difficult, as the spatial resolution of the training data is of the lower order of magnitude. Still some linkage can be found. Adams and Kanaroglou (2016), Krecl et al. (2019) and Van den Bossche et al. (2018) used mobile air quality measurements to train their models to produce spatial air quality predictions and concluded that the models tend to underestimate the localized peak values, as also shown in this study. Peng et al. (2017) also found that linear models lead to a smaller bias for a short data record when applying modelled air quality data to conduct up
to 48 h air quality predictions. On the contrary, SVR was shown to outperform all other methods in Hu et al. (2017). However, their data had a spatial resolution of 15 m or more, which can explain why the other models had trouble generalizing the data. To achieve the best generalization and to avoid over-fitting, Van den Bossche et al. (2018) stresses the importance of high quality model predictors, high temporal and spatial resolution of training data and evaluation against an independent dataset, which are all fulfilled in this study.
A downside of the developed models is that, when applied to a new city plan, they require the plan to lead to a similar statistical distribution of the pollutant concentrations as the ones used for training, which means that the models have to cope with moderate amounts of concept drift. We can however detect this to avoid potentially false predictions that may occur when applying the models on data that does not follow the training distribution. Yet another limitation is the small number of





simulation set-ups of the data: there are only eight different simulation set-ups (i.e., four city plans and two wind direction)
to train with, which rules out methods requiring large annotated training datasets, such as some applications of deep learning.
With more data the models could potentially reach completely another level of accuracy. Another way to improve the accuracy
would be to have even better and more capturing features. If very accurate results are needed, running a new LES is still the best
approach. Still, model predictions are accurate enough for many purposes, such as to study pollutant exposure and to support
urban planning.

The developed methods can be used to further probe new meteorological conditions and city plans. In a future study, these
models could help us to understand how simple changes into the layout of the city plan, e.g., a new building, affect the local
air pollutant concentrations. Eventually similar models could also be used to understand the complicated phenomena in simple
urban areas.

To conclude, we have explored using different machine learning models how to emulate air pollutant concentrations as
simulated using LES. We use LES made over two different boulevard-type street canyons to study the impact of building-block
layouts on air pollutant concentrations. We examine the performance of ten machine learning methods by using site-specific
features to predict the surface-level concentrations over the neighborhoods. A total of 20 features are determined from the
LES inputs and outputs. The results show how the studied machine learning methods are able to produce the mean pollutant
concentrations. Further, concept drift detection is used to detect areas where the model cannot be trusted and more simulation
runs may be needed.

*Code and data availability.* The code to reproduce the regression models is available at
http://doi.org/10.5281/zenodo.3999302. The input and output data for KU18 are available at
http://urn.fi/urn:nbn:fi:att:cfe1bd77-6697-44b5-bdd7-ee74f36c7dcd. The input data for KA20 are available at
http://doi.org/10.5281/zenodo.3556287 and the output data will be made available at
http://urn.fi/urn:nbn:fi:att:ee275362-3f56-477c-bbbc-6fcacd9c7f95.



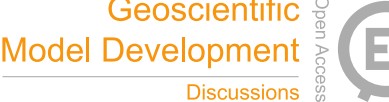

# Appendix A

Table A1 contains detailed descriptions of the features used in the regression models.

| Feature | Description |
| --- | --- |
| Building height | Height over ground at the nearest point of the nearest building. |
| Canopy height | Height of the vegetation canopy at point $(x,y)$. |
| Courtyard | Equal to 1 if a there exists a cross laid on a point $(x,y)$ that intersects the same building on all four sides, else 0. |
| Direction of closest building | Direction of the closest building with respect to the mean wind direction. Due to symmetry has values between 0 and $\pi$ (= 180°). |
| Distance to building downwind | Distance to the next building when walking directly with the mean wind direction. Not defined in areas that do not have buildings in the direction of the mean wind. |
| Distance to building upwind | Same as 'Distance to building downwind', when walking *against* the mean wind direction. |
| Height to width ratio | Height to width ratio of the area that the point is in (e.g., a street), calculated as 'Building height' divided by 'Street width'. |
| Pollutant emissions | Pollutant emission factor weighed by the street type (see Kurppa et al. (2018) for details). Takes values 0, 1, 2, 4. |
| Pollutant emissions convolution, $\sigma = \{1, 2, 4, 8, 16\}$ | Weighted pollutant emissions in the surroundings. Convolution of the weighted pollutant emissions, using a Gaussian kernel of varying sizes $\sigma$. |
| Pollutant emissions convolution upwind, $\sigma = \{8, 16, 32\}$ | Same as 'Pollutant emissions convolution', when walking against the mean wind direction. This is estimated as a cone that covers a triangular area in the upwind direction. The kernel is a normalized cut-out of an ordinary Gaussian kernel, that resembles a cone pointing away from the mean wind source. |
| Street | Equal to 1 if point $(x,y)$ is in an inhabited area (is_inhabited = 1), is not a courtyard or an intersection (is_courtyard = is_intersection = 0), and has a street width of less than 100 (street_width < 100), else 0. |
| Street width | The sum of the distance from point $(x,y)$ to the nearest building, plus the distance from point $(x,y)$ to second nearest building. For courtyards, the minimum is taken (e.g. street width in a courtyard sized 10m x 20m is defined to be 10m). |

**Table A1.** Detailed descriptions of features in Table 1. The supplementary material contains visualisations of these features. Note that these features were computed with the help of auxiliary features that are not listed here, such as is_building, is_inhabited, and is_intersection. For more details, see the supplied software code.



*Author contributions.* LJ, MK, KP and EO designed the concept of the study. ML and HM pre-processed the LES outputs, prepared and conducted the machine learning simulations and statistical analyses with contributions from EO, RS and KP. LJ and MK provided expert

advice on the LES model inputs and outputs. All co-authors participated in writing the manuscript with contributions from all co-authors.

*Competing interests.* The authors declare that they have no conflict of interest.

*Acknowledgements.* For financial support we would like to thank the Academy of Finland (profiling action 3 and decisions 326280 and 326339), Helsinki Institute for Information Technology HIIT, the Doctoral Programme in Atmospheric Sciences (ATM-DP), and the Doctoral Programme in Computer Science (DoCS) at the University of Helsinki, and SMart URBan Solutions for air quality, disasters and city growth

(SMURBS, no. 689443) funded by ERA-NET-Cofund project under ERA-PLANET.



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
