# Peer review of "Machine learning models to replicate large-eddy simulations of air pollutant concentrations along boulevard-type streets - Supplementary material"

_Geoscientific Model Development, 2020_

## Referee Comment (RC1) · Anonymous Referee #1 · 22 Mar 2021

General Comments

Lange and coauthors do an admirable job of presenting a difficult concept to make clear, that is, their algorithm is essentially a model of a model. The study makes clear the benefits of such an absurd sounding task, that the CFD models in question are computationally very expensive. And so the authors present a much faster algorithm which produces similar results, with the intent of providing an easy-access tool for first-pass examination of urban planning on local-scale air quality. The authors were very thorough in their assessment of the available algorithms and justify their choices. There are a lot of open questions over how to improve the model performance in the street

canyon, and these are only highlighted by the Drift detection discussion, but I believe these can be left to future work. Overall I think the manuscript is worthy of publication.

Specific Comments

I have one minor technical concern: the use of a scaling variable feels a bit suspect on first reading. After a closer reading, it feels to me like a presentation problem. That is, the scaling variable is presented as something apart from the model when it really is a part of the model, since it will be needed to make future predictions. I would prefer to see a discussion of how the scaling variable is chosen for the cross-validation / evaluation cases: is it simply the value found in the minimization of RMSE on the training data carried forward, or are the authors recomputing the scaling variable for the evaluation data? I suspect the former, and I would like to see that confirmed in the methods section 2.5. If the latter, then there needs to be more description of this process and how the authors intend to compute the multiplicative factor a for novel datasets.

Technical Corrections

The paper is very well written and there were no glaring grammatical errors.

---

## Referee Comment (RC2) · Anonymous Referee #2 · 15 May 2021

This is a very well-conducted study and in my opinion, it is warranted for publication for the following reasons: 1) the main idea (regressive model identifying useful features for time-averaged pollutant prediction on 2d grid from LES data) is sound. 2) The methodology is well defined. The reasononing behind the choice of train and test datasets are explained. A well-established feature selection method is used. The pool of candidate features is also extensive. 3) The "concept detection" in section 3.4 is another interesting check on the validity of model.

Some comments for the authors to think about and address:

1) The issue with the negative output of the models is puzzling. The authors scale input features. Why not use a scaling of the target variable such that the output data is projected between [-min, max]. Every time the model is queried, the output of the model can go through the inverse transform of the scaling to scale it back to physical values. Could this solve the clipping of the output problem?

2) SVR results seem pretty reasonable too. In fact, they have equal num features with log-norm SVR and smaller bias and RMSE on PM2.5. Is there a reason the authors have chosen not to use/highlight that model

3) The dummy model while being a simple reference (no feature selection needed, no training, etc.). It is too much "dummied down" in my opinion to draw comparisons against, especially in the conclusions and abstract. My suggestion is to take the linear reg as the baseline. Not much would change as your three highlighted models still perform better in at least one sense (fewer features, better RMSE on one of the two tests). This would be a stronger comparison and reflects better on your results.

---

## Author Comment (AC1) · 4 Jun 2021

We thank the referees for their encouraging comments.

Regarding the specific comments by the referees:

RC1, scaling variable: The reason for the scaling variable is, as stated in the manuscript (page 9), that the scaling - including units of measurements - of air pollutant concentrations in the evaluation data (KA20) differ from those in the training data (KU18). For this reason, we use an error measure (mRMSE) that only depends on the relative "shape" of the pollutant concentration and that is invariant to linear scaling

of the concentrations in the training and/or evaluation data. Because the scaling of the training and evaluation data differ it is not possible to compute this scaling factor by using the training data alone, which is why we compute the scaling variable on the evaluation data (KA20) by finding the scaling which minimises the RMSE on the evaluation data.

For new evaluation data set we could either use the same multiplicative constant (if the scaling in the new evaluation data set is expected to be identical the scaling in the old evaluation dataset) or find a new multiplicative constant.

(An alternative interpretation would be that the scaling variable would be part of our regression model and that we would use the standard RMSE error measure, in which case we would make a (small) error when we find the scaling constant by using the evaluation data. Even with this alternative interpretation, the resulting overfitting should however not be substantial, because we are fitting here only one number to the evaluation data.)

We will add a short discussion about these issues to the description of mRMSE error measure.

RC2, comment 1: The reviewer is correct in pointing out that we could solve the problem of negative outputs, e.g., in the simple linear models by transforming the output variable to positive values. In fact, this is what is effectively accomplished by the Poisson and log-linear regression models which transform the linear response, e.g., by using exponential transformation to non-negative domain and which do not therefore have the problem of negative outputs. We have included the linear model in the manuscript, because we just wanted to include "naive" linear regression as a simple baseline model. We will clarify this in the final manuscript.

RC2, comment 2: As pointed out in page 11 of the manuscript, standard SVR leads to practically similar performance with logarithmic SVR. Out of these two SVR models we choose to highlight the latter (logarithmic SVR) because it has always non-negative

predictions and because the performance and chosen features are otherwise practically identical with the standard SVR.

RC2, comment 3: Reporting the RMSE for the dummy model allows to assess the overall benefit of using a regression model at all. For example, knowing (m)RMSE for the dummy model allows to easily estimate the commonly used $R^2$ metric (e.g., $R^2 = 1 - 0.76^2/1.78^2 = 0.82$ for the log-linear regression mentioned in the abstract). For these reasons, we feel that reporting the dummy model RMSE as a baseline is informative.

Sincerely,

The Authors

---

## Author Response (AR1)

Dear editors,

We have now taken the referee comments into account. Please find below summary of the changes.

We refer to our author comment at https://doi.org/10.5194/gmd-2020-200-AC1 for our detailed response to the referees' comments. Below, we summarize changes we made to the manuscript based on these comments. Additionally, we updated some of the references.

The referees had a total of 4 comments (1 by referee RC1, 3 by referee RC2).

*Comment by RC1 about scaling variable:*

We added the following paragraph to Sect. 2.5 on page 9 to clarify the issue raised by the reviewer:

**mRMSE therefore depends only on the relative magnitudes of the pollutant concentrations and it is invariant to linear scaling of the training or evaluation data. For a new evaluation dataset, we could either use the same multiplicative constant -- if the scaling in the new evaluation dataset is expected to be identical to the scaling in the old evaluation data -- or find a new multiplicative constant.**

**Comment by RC2 about the problem of negative outputs (comment 1):**

We added the following clarifying sentence to Sect. 3.1 on page 10 (our addition has been **bolded** here):

**A possible** approach for avoiding negative predictions is **either to use a transformation that allows only positive predictions (for example, log-linear regression instead of linear regression), or** to clip model outputs to a minimum of zero after prediction.

**Comment by RC2 about logarithmic vs. standard SVR (comment 2):**

As argued in the author comment, we feel that we have already covered this issue in Sect. 3.2 on page 11. We therefore did not modify the manuscript.

**Comment by RC2 about choice of the dummy model (comment 3):**

As argued in the author comment, we feel that including the dummy model as a baseline is informative. We therefore did not modify the manuscript.

Yours sincerely,

Kai Puolamäki, on behalf of the authors